# Basalt Fibre Composite with Carbon Nanomodified Epoxy Matrix under Hydrothermal Ageing

**DOI:** 10.3390/polym13040532

**Published:** 2021-02-11

**Authors:** Tatjana Glaskova-Kuzmina, Aldobenedetto Zotti, Anna Borriello, Mauro Zarrelli, Andrey Aniskevich

**Affiliations:** 1Institute for Mechanics of Materials, University of Latvia, 3-635, Jelgavas Str., LV-1004 Riga, Latvia; Andrey.Aniskevich@pmi.lu.lv; 2Institute of Polymers, Composites and Biomaterials, National Research Council of Italy, 80055 Portici, Italy; aldobenedetto.zotti@unina.it (A.Z.); anna.borriello@cnr.it (A.B.); mauro.zarrelli@cnr.it (M.Z.)

**Keywords:** polymer composite, carbon nanofiller, epoxy, hydrothermal ageing, mechanical properties, water absorption, electrical resistance

## Abstract

This work aimed to investigate the effect of hybrid carbon nanofillers (e.g., carbon nanotubes/carbon nanofibers in the ratio 1:1 by mass) over the electrical and flexural properties for an epoxy matrix and corresponding basalt fibre reinforcing composite (BFRC) subjected to full-year seasonal water absorption. Hydrothermal ageing was performed by full immersion of the tested materials into distilled water according to the following model conditions (seasons). The mechanical properties were measured in three-point bending mode before environmental ageing and after each season. Upon environmental ageing, the relative change of flexural strength and elastic modulus of the epoxy and NC was within 10–15%. For nanomodified BFRCs, the slightly higher effect (approx. by 10%) of absorbed moisture on flexural characteristics was found and likely attributed to higher defectiveness (e.g., porosity, the formation of agglomerates etc.). During flexural tests, electrical resistance of the nanocomposites (NC) and BFRC/NC samples was evaluated. The electrical conductivity for UD BFRC/NC, before and after hydrothermal ageing, was by 2 and 3 times higher than for the NC, accordingly, revealing the orientation of electrically conductive nanoparticles and/or their agglomerates during lay-up manufacturing which was evaluated by the rules of the mixture. Based on all results obtained it can be concluded that the most potentially applicable for damage indication was UD BFRC/NC along fibres since full-year hydrothermal ageing improved its electrical conductivity by approx. 98% and, consequently, the ability to monitor damages was also enhanced.

## 1. Introduction

Environmental ageing is one of the most important reasons for the preliminary failure of fibre reinforced plastics (FRP) applied in engineering structures for outdoor applications [1,2]. Basalt fibre reinforced composites (BFRCs) are sustainable materials that can be positioned between glass fibre reinforced composites (GFRCs) and carbon fibre reinforced composites (CFRCs) in terms of performance and cost-effectiveness. Besides, BFRCs also have high stability to the action of water, acids, and chemically active media which results in prolonged durability of FRPs during operation in the outdoor environments in comparison with CFRCs and GFRCs [1,3,4]. Based on these advantages, the applicability of BFRCs as structural materials for outdoor applications is highly expected [5,6].

Nevertheless, the durability of FRP composites is a complex phenomenon and it depends on the degradation of both polymer matrix and fibres, as well as on their interface bond behaviour [7,8,9]. The epoxy resins which are usually applied for the impregnation of FRPs may have changes in their physical and chemical properties due to environmental effects (e.g., moisture, temperature, and UV radiation) [10,11,12] leading to plasticization of the epoxy resins, chemical reactions, and post-curing.

The addition of homogeneously dispersed carbon nanofillers such as nanotubes (CNT), nanofibres (CNF), and graphene and could lead to environmental stability enhancement of mechanical, thermal, and electrical properties [8,11,13,14,15]. Besides, it may introduce additional functionality to nonconductive polymer resins and FRPs [14] and it provides structural health monitoring, i.e., the control of mechanical properties through the changes in electrical properties. Nevertheless, the knowledge of durability and environmental ageing of such advanced composites is still under investigation limiting their potential applications.

Based on previously obtained experimental and theoretical results on accelerated hydrothermal ageing [11], the epoxy filled with 0.1 wt.% of hybrid nanofiller, CNT and CNF in the ratio 1:1 by weight, was characterized by the lowest effect of both temperature and moisture on material characteristics, together with the lowest sorption characteristics, indicating improved stability to the environmental factors. The novelty of the current paper is the implementation and comprehensive analysis of the hydrothermal ageing for the neat, nanomodified epoxy and BFRC, by using previously obtained findings for the most environmentally stable NC.

This work aimed to investigate the effect of the hybrid nanofiller (HN) over the electrical and mechanical properties for an epoxy matrix and BFRC laminates subjected to full-year seasonal water absorption. Thus, the first task of the aim was to evaluate the mechanical performance of nano-modified BFRCs in such seasonal conditions. The effect of absorbed water on the flexural characteristics was analysed in bending since polymer matrix exhibits a major effect in this loading mode for the mechanical properties of FRPs if compared to tension or compression. The flexural characteristics of the nano-modified BFRCs were compared with corresponding unmodified composites and with nano-modified epoxy undergone to the same model conditioning.

The second task of the aim was to establish the effect of hydrothermal ageing on the electrical resistivity/conductivity revealing if the ability to indicate damages remains. Therefore, for all conductive test samples, the electrical resistivity was measured during three-point bending tests before and after the hydrothermal ageing.

## 2. Materials and Methods

A commercially available monocomponent RTM6 (Hexcel Composites, Stamford, Connecticut, USA) epoxy resin was used as a matrix system. Multiwall carbon nanotubes (CNT) Nanocyl 7000 (Nanocyl, Sambreville, Belgium) and carbon nanofibers (CNF) SA 719781 (Sigma-Aldrich, St. Louis, MO, USA) were chosen to prepare hybrid nanofiller and used as received. Previously, it was verified that the addition of such hybrid nanofiller (in the ratio 1:1 by mass) at the content of 0.1 wt.% improved hydrothermal resistivity for the epoxy and electrical conductivity of approx. 0.01 S/m which was enough for structural health monitoring [11,16]. Unidirectional (UD) basalt fabric BAS UNI 350 (Basaltex, Wevelgem, Belgium) was used to prepare neat and nano-modified BFRCs.

The dispersion of hybrid nanofiller within the epoxy matrix was performed by high-shear Ultra Turrax T25 disperser combined with a heating plate RCT basic (IKA, Staufen im Breisgau, Germany). The dispersion procedure and preparation of NC test samples were previously described in [11] reporting an optimized final mixture. After preliminary microstructural analysis of the dispersion of hybrid nanofiller in the epoxy resin by using optical microscopy reported in [11] the bimodal distribution by filler particle area with characteristic peaks at approx. 0.2 and 2.0 µm^2^ was revealed for both single and hybrid nanofillers. The appearance of two peaks for area distribution was attributed to the existence of two characteristic sizes of the filler particles: separated filler particles of radius approx. 250 nm and their associated agglomerates of size approx. 790 nm.

For the preparation of BFRC/epoxy and BFRC/NC laminates the lay-up technology by using vacuum-assisted heating plate was used. Unidirectional lay-up orientation [0]_8_ was considered and related BFRC samples were manufactured. For the laminates, the test samples were cut in the direction parallel (par) and perpendicular (perp) to the fibres. Thus, BFRC/epoxy and BFRC/NC in the direction parallel to basalt fibres are denoted as “U par” and “UN par”, while BFRC/epoxy and BFRC/NC in the direction perpendicular to fibres—“U perp” and “UN perp”, accordingly.

The sizes of the samples were approx. 125.0 mm × 10.0 mm × 2.5 mm for the laminates and 80 mm × 10 mm × 3.5 mm for the epoxy and NC. The samples were cut manually and polished before testing to reduce the effect of surface roughness on the kinetics of water sorption. The number of replicates was five for all tests and materials investigated and the values shown on the figures represent the average value together with its standard deviation shown as error bars.

The process of full-year hydrothermal ageing was simulated by full immersion in distilled water following four model seasons corresponding to approx. climatic conditions of Northern Europe: 3 months of summer (20 °C), autumn (10 °C), winter (−10 °C), and spring (10 °C). The tested specimens were periodically weighed by using XS205DU balance (Mettler Toledo, Columbus, OH, USA) with a precision of 0.05 mg. Water absorption tests at room temperature until equilibrium were performed for all test samples to obtain full kinetics of moisture sorption.

The mechanical properties were tested in three-point bending according to ASTM D790 before the environmental ageing and after each season. The support span of 56 mm and a strain rate of 1.5 mm/min were chosen and applied for the test samples by using Zwick 2.5 machine (Zwick Roell Group, Ulm, Germany). From the stress-strain curves, the elastic modulus and flexural strength were evaluated. The electrical resistance of the NC and BFRC/NC samples was measured by using two-point methodology by using a multimeter DMM 4020 (Tektronix, Beaverton, OR, USA) with a precision of 1 kOhm. Opposite facets of the samples were covered with conductive silver paint to reduce contact resistance effect.

## 3. Results and Discussion

### 3.1. Porosity Analysis

The relative volume of pores (i.e., open pores, voids etc. defects on the surface of the test samples) of the epoxy, NC, BFRC/epoxy and BFRC/NC was evaluated by using hydrostatic weighing in distilled water by using XS205DU balance (Mettler Toledo, Columbus, OH, USA):(1)$$v_{p}=\frac{V_{p}}{V},$$
where *V_p_* and *V* is the volume of the pores and the total volume, and:(2)$$V_{p}=V-\frac{m_{a}-m_{l}}{\rho_{l}},$$
while the total volume of the samples *V* was found as the multiplication of the geometrical sizes of the sample, *m_a_* and *m_l_* were the mass of the sample in the air and the liquid, and *ρ_l_* was the density of the liquid used for the hydrostatic weighing.

The results obtained for the relative volume of pores of all tested materials are provided in Figure 1. According to Figure 1, the relative volume of open pores, voids and other defects located on the surface of the test samples for the epoxy was 2–3 times smaller in comparison with the other investigated materials. Likely, it can be attributed to the lowest impact of the manufacturing conditions since the dispersion of carbon nanofillers and/or further using of vacuum-assisted resin transfer in the lay-ups of the laminates might introduce additional manufacturing defects, pores and/or delaminations within the materials. It should be noted that the existence of such defects could have negative effects on the mechanical and electrical properties (only for the conductive systems) and they may lead to the enhanced moisture uptake during environmental ageing and subsequently enhanced degradation of the materials.

### 3.2. Water Absorption

During hydrothermal ageing, all test samples were fully immersed in distilled water and periodically weighed at room temperature and after each season. For the description of experimental data for water absorption in the epoxy and NC, which can be regarded as isotropic media, the three-dimensional Fick’s model is usually used [9,14,17,18] as follows:(3)$$w(t)=w_{\infty }-\left(w_{\infty }-w_{0}\right)\frac{8}{\pi^{6}}\sum_{k=1}^{\infty }\sum_{n=1}^{\infty }\sum_{m=1}^{\infty }\frac{{\left[1-{\left(-1\right)}^{k}\right]}^{2}{\left[1-{\left(-1\right)}^{n}\right]}^{2}{\left[1-{\left(-1\right)}^{m}\right]}^{2}}{k^{2}n^{2}m^{2}}\exp\left[-\lambda^{2}{}_{k,n,m}Dt\right],$$where $\lambda_{k,n,m}^{2}=\lambda_{k}^{2}+\lambda_{n}^{2}+\lambda_{m}^{2}={\left(\frac{\pi k}{a}\right)}^{2}+{\left(\frac{\pi n}{b}\right)}^{2}+{\left(\frac{\pi m}{l}\right)}^{2}$; *w*_0_ and *w*_∞_ are the initial and equilibrium moisture contents; and *a*, *b*, and *l* are the thickness, width, and length of the specimens. *D* is the diffusion coefficient of the material.

The equilibrium moisture content was found as the maximal value for the moisture content obtained by the materials during water absorption tests. The diffusion coefficient of the epoxy and NC was obtained by calculating it from the initial slope of the curve *w* ($\sqrt{t}$) [19]:(4)$$D=\frac{\pi h^{2}}{16t}{\left(\frac{w(t)-w_{0}}{w_{\infty }-w_{0}}\right)}^{2}.$$

For the description of water absorption in unidirectional BFRCs the samples which are anisotropic media, Equation (3) should be modified, accordingly:(5)$$w(t)=w_{\infty }-\left(w_{\infty }-w_{0}\right)\frac{8}{\pi^{6}}\sum_{k=1}^{\infty }\sum_{n=1}^{\infty }\sum_{m=1}^{\infty }\frac{{\left[1-{\left(-1\right)}^{k}\right]}^{2}{\left[1-{\left(-1\right)}^{n}\right]}^{2}{\left[1-{\left(-1\right)}^{m}\right]}^{2}}{k^{2}n^{2}m^{2}}\exp\left[-\lambda_{k,n,m}^{2}t\right],$$where $\lambda_{k,n,m}^{2}=\lambda_{k}^{2}+\lambda_{n}^{2}+\lambda_{m}^{2}=D_{22}{\left(\frac{\pi k}{a}\right)}^{2}+D_{22}{\left(\frac{\pi n}{b}\right)}^{2}+D_{11}{\left(\frac{\pi m}{l}\right)}^{2}$ for BFRC/epoxy and BFRC/NC samples cut in the direction parallel to the orientation of the fibres (U par and UN par), and $\lambda_{k,n,m}^{2}=\lambda_{k}^{2}+\lambda_{n}^{2}+\lambda_{m}^{2}=D_{22}{\left(\frac{\pi k}{a}\right)}^{2}+D_{11}{\left(\frac{\pi n}{b}\right)}^{2}+D_{22}{\left(\frac{\pi m}{l}\right)}^{2}$ for the samples in the direction perpendicular to the orientation of the fibres (U perp and UN perp).

Many fillers like glass, carbon and basalt fibres as well as mineral additives are hydrophobic or nearly hydrophobic, and the diffusion coefficient of the fibres is much smaller than corresponding value for the polymer matrix. In this case, it is assumed that the *D* value of the fibres is close to zero, thus BFRCs parallel, *D*_11_, and normal, *D*_22_, to the fibres have been computed by using the following relationship [9,19]:(6)$$D_{11}=D_{m}\left(1-v_{f}\right)$$
(7)$$D_{22}=D_{m}\left(1-2\sqrt{v_{f}/\pi }\right)$$
where *D_m_* is the diffusion coefficient of the matrix and *v_f_* is the volume fraction of the fibres.

Assuming mass fraction of the fibres, densities of epoxy resin and basalt fibres, respectively, equal to *c_f_* = 69.7%, *ρ_m_* = 1.14 g/cm^3^ and *ρ_f_* = 3.00 g/cm^3^, it was found that the volume fraction of fibres, *v_f_*, for all BFRC systems was estimated about 46.6% [20], according to the following equation:(8)$$v_{f}=\frac{\rho_{m}\cdot c_{f}}{\rho_{m}\cdot c_{f}+\rho_{f}\cdot \left(1-c_{f}\right)}$$

The water absorption kinetics for all tested materials is provided in Figure 2. For the epoxy and NC system, being isotropic material, the Fick’s model has provided an effective description (Figure 2a) however, it was less representative of the acquired data for BFRCs (e.g., an orthotropic system) in the parallel (Figure 2b) and perpendicular (Figure 2c) fibre direction. Notably, the sorption process in the epoxy and NC was less complicated than in BFRCs, resulting in much smaller data scattering and thus leading to a better description by using Fick’s model. Moreover, it should be noted that the sorption process reached the equilibrium state, only in the case of epoxy and NC, whereas it is still progressing for BFRC samples. For this reason, the BFRC equilibrium moisture content, *w*_∞_, used in Equation (5), was computed taking into account the experimental data obtained in the case of water absorption until equilibrium at elevated temperatures (50 and 70 °C), not reported within this manuscript. The equilibrium water content, at both temperatures, was almost the same (2.67 ± 0.02% for U par, 2.65 ± 0.02 for UN par, 2.28 ± 0.01% for U perp, and 2.27 ± 0.02% for UN perp) and, thus, it was used for the calculation at room temperature until equilibrium, nevertheless the sorption process is still in progress.

For the quantitative analysis of the sorption behaviour of all tested materials, their sorption characteristics, i.e., diffusion coefficient and equilibrium water content, are summarized in Figure 3. It was found that the diffusion coefficient of the NC was slightly higher than for the epoxy although, for BFRCs nano-modified systems (i.e., BFRC/NC (22)) its minimum value was recorded. The reason for such results could be a higher content of defects (e.g., pores, delaminations, etc.) introduced during the manufacturing process, this augmentation is supported by the volume content estimation of open pores and voids on the surface of the tested samples as shown in Figure 1.

According to Figure 3, BFRCs nano-modified systems were characterized by lower values of diffusion coefficient both in parallel and perpendicular direction of the fibres. Such reduction likely due to addition of carbon nanofillers could be explained by assuming a real decrease of the free volume available due to moisture absorption and reduced permeability, being a function of the nanoparticles volume fraction and aspect ratio [8,11,13,14,18]. Moreover, from Figure 3 it can be noted that the equilibrium water content, in the case of BFRCs, is higher compared to those for the epoxy and NC and this could be related to the increased porosity of these systems in comparison with the epoxy allowing water molecules to fill the free volume and pores inside the material.

Furthermore, according to Figure 2, the control data of water content registered after each season is following the same route and is located almost on the calculation curve obtained for water absorption at room temperature (approx. 20 °C). It could be explained by the fact that in 3 months of the summer season which also performed at room temperature the test samples have absorbed about 70–80% of the maximal water content and further water uptake at lower temperatures didn’t contribute much to the kinetics of water absorption.

### 3.3. Flexural Properties

The mechanical properties were evaluated in three-point bending to investigate mainly the effect of absorbed water on epoxy and NC which were used also for the impregnation of BFRCs. The procedure to evaluate flexural stress, strain, elastic modulus and strength is a standard one and was previously described in details in [11]. The representative stress-strain curves before the seasonal hydrothermal ageing are provided in Figure 4a for the epoxy and NC and in Figure 4b for all BFRC systems. According to Figure 4, no significant effect of the addition of hybrid nanofiller was observed for all materials. It may be reasonably associated with the negligibly small nanofiller content—only 0.1 wt.% in the NC and 0.03 wt.% in BFRC. Notably, the addition of such small nanofiller content didn’t improve the mechanical characteristics such as elastic modulus and strength. On the other hand, almost the same mechanical properties for neat and nano-modified materials proved that the amount of the defects introduced during the manufacturing process (due to already mentioned increased porosity) did not change significantly the mechanical properties at least in the unconditioned state (i.e., before the hydrothermal ageing).

The same mechanical testing was performed after each season, and the flexural characteristics were measured for all materials as a function of absorbed water content and they are provided in Figure 5 which clearly shows that the effect of absorbed water on the flexural strength and elastic modulus of the epoxy and NC was not significant. The relative change of these characteristics was within 10–15%. Winter was one of the most important seasons causing brittleness of the materials and, as a result, a slight decrease of both elastic modulus and flexural strength for all materials tested.

Similar results for the elastic modulus and flexural strength were obtained also for BFRC systems, which are summarized in Figure 6. Notably, absorbed moisture did not cause plasticization of the tested materials since both mechanical characteristics negligibly changed after full-year hydrothermal ageing. According to Figure 6, for nano-modified BFRCs, the slightly higher effect of absorbed moisture on these characteristics was found and it is likely attributed to higher defectiveness (e.g., porosity, the formation of agglomerates etc.). Also, it may be noted from Figure 6 that the amount of absorbed water for nano-modified BFRC systems was reduced if compared to BRRP/epoxy systems. This result could be attributed to a high aspect ratio of CNF and CNT, i.e., approx. 150, which could contribute to an increased tortuosity of the path for the water molecules during moisture diffusion as well as a restriction of the molecular dynamics for the polymer chains around the nanoparticles, causing retardation of their relaxation.

Usually, it is assumed that fibres in FRPs are less influenced by the environmental factors and the moisture absorbed by polymers and polymer-based composites mostly results in swelling, plasticization and acceleration of physical ageing of the polymers. Nevertheless, the shrinkage of fibres and hydrolytic breakdown of fibre-matrix interface leading to a loss of efficiency of stress transfer at fibre/matrix interface [21]. For the case of nano-modified FRPs, the incorporation of the additional interfacial area between carbon nanofiller particles and polymer matrix exists which is expected to activate the additional energy dissipation mechanisms related to the interfacial sliding, fibre pull-out, and bridging, as well as crack bifurcation and retardation at the nanoscale [22].

Moreover, due to the difference in swelling coefficients of the matrix and filler/fibre and non-uniform moisture concentration, large stresses can appear in polymer-based composites, especially at the interfacial area causing its weakening [11,23]. Therefore, minor changes in the flexural characteristics both for the epoxy and BFRCs systems during full-year hydrostatic ageing indirectly testified that no significant plasticization occurred, and the investigated materials could be regarded as environmentally stable materials.

### 3.4. Electrical Properties

First, the electrical resistivity was measured for all conductive materials tested before the hydrothermal ageing and during three-point bending tests to find the correlation between the flexural strain and resistivity. Plenty of results were reported in the literature on the improvement of electrical conductivity in carbon nanofiller polymer systems which was mainly explained by the high values of electrical conductivity and aspect ratio of the carbon nanofillers along with the ability to form percolative networks at very low filler content [22,24,25,26,27]. In the current study, the relative change in the electrical resistance during mechanical testing was calculated as:(9)$$\frac{\Delta R(t)}{R_{0}}\left(\%\right)=\frac{R(t)-R_{0}}{R_{0}}\cdot 100,$$

The results obtained for the relative change of electrical resistance of all conductive materials are provided in Figure 7 as a function of the flexural strain. Obviously, for all materials, the positive piezoresistive response was obtained when the change in strain caused a positive change in the electrical resistance. For the case of BFRC/NC in the direction parallel (UN par) and perpendicular (UN perp) to fibres the relative change of electrical resistance was much higher (approx. 6%) than for the NC (approx. 1%). Generally, for all materials, such a response can be attributed to the change of CNTs’ intrinsic resistance subjected to the applied strain and intertube resistance resulted from the change of CNT network structure [24].

Moreover, the variation in electrical conductivity/resistivity of such multifunctional composites could be directly related to the macroscopic strain experienced by the material and the internal damages which appear during the lifetime of the materials. Two co-directional phenomena for the increase of electrical resistance due to tensile load were reported in [24,27]: change in the configuration in CNT conductive network and geometrical effect of dimensional changes of a sample. The compressive loading vice versa resulted in a decrease of electrical resistance of polymer resins filled with CNTs which was attributed to the decrease in gaps between CNTs and/or their associated agglomerates building the conductive network being pushed together [28,29].

Three-point bending tests involve both tensile and compressive internal stresses acting simultaneously on the sample and, thus, they are more complicated for the analysis. For this reason, since the total resistance of all tested materials increased during three-point bending tests, based on results provided in Figure 7, it may be concluded for all systems the effect of tensile loads was greater than compressive loads, and the intertube resistance might mainly increase due to the variation of tube-tube contact.

As many physical properties, also the electrical conductivity/resistivity of electropassive materials may be influenced by the effect of environmental factors such as moisture and temperature. Hydrothermal exposure generally reduces electrical resistance [22,27] due to the presence of water in the materials especially when the water is located in pores and microvoids. Moreover, absorbed water molecules can interact with the polymer forming physical and/or chemical sites, forming clusters and causing regrouping of water molecules triggering ionic conduction mechanism [27].

The effective electrical conductivity of the materials is the inverse of electrical resistivity and it was calculated as follows:(10)$$s=\frac{l}{R\cdot A},$$
where *l* is the length between the specimen facets (specimens’ length) where the conductive paint was applied, and *A* is the cross-section area of the specimen.

Surely, the degree of such effects could be varying for different materials. The results obtained for the electrical conductivity of all tested materials before and after full-year environmental ageing are summarized in Figure 8. It is interesting to note that for BFRC the electrical conductivity, in the direction perpendicular to fibres, remained the same while for NC and, particularly for BFRC along the fibres, increased significantly.

The increase of NC electrical conductivity provided in Figure 8, after the hydrothermal ageing, can be attributed to the redistribution of water molecules leading to ionic conduction, which, in turn, contributes to the total conductivity of the nanocomposite. The significant increase in the electrical conductivity of BFRC in the direction parallel to fibres could be addressed to the effects of the orientation of hybrid carbon nanofiller during the manufacturing process since the same NC content was applied manually for each ply. To prove this hypothesis, the electrical conductivities of BFRC in the direction parallel and perpendicular to fibres were estimated before and after hydrothermal ageing by using modified rules of mixture (assuming that basalt fibres are not electrically conductive) [30]:(11)$$s_{11}=n\cdot s^{NC}\cdot (1-v_{f}),$$
(12)$$s_{22}=s_{33}=\frac{1}{n}\cdot s^{NC}\cdot \left(1-\frac{v_{f}}{2v_{f}+\frac{1}{2}}\right),$$
where *n* is the factor attributed to carbon nanofiller orientation, $s^{NC}$ is the electrical conductivity of NC before (0.0062 ± 0.0010 S/m) or after (0.0080 ± 0.0015 S/m) the hydrothermal ageing which was obtained experimentally, *v_f_* (0.47) is the volume content of fibres, and *s*_22_ = *s*_33_ due to fibre orientation along axis 1.

The results of the evaluation of electrical conductivity of BFRC along (UN par) and perpendicular (UN perp) to the fibres by using Equations (11) and (12), accordingly, is provided in Figure 9 together with experimental data. Notably, by using factor n which is indicating on the enhanced (*s*_11_) or reduced (*s*_22_) orientation of hybrid carbon nanofiller it is possible to fit the experimental results both before and after hydrothermal ageing as well as for both lay-up orientations. According to Figure 9, the factor of 5 (five times higher electrical conductivity of NC than experimentally obtained) gives the most appropriate fitting for BFRC in the direction along the fibre orientation, while the factor of 3 (three times lower electrical conductivity of NC than experimentally obtained) provides the best fitting for BFRC in the direction perpendicular to fibre orientation. Thus, the hypothesis regarding the possible orientation of particles of hybrid carbon nanofiller during the manufacturing process was proved. Of course, such estimation is being made by assuming the rest factors nonsignificant and non-contributing to the overall electrical conductivity of the BFRC.

Based on results obtained, the most potential system, suitable to monitor damage evaluation was the UD BFRC/NC along fibres since full-year hydrothermal ageing improves its electrical conductivity and, consequently, its sensitivity to health diagnostic was enhanced. The addition of only 0.03 wt.% of hybrid nanofiller to BFRCs allowed the piezoresistive response to be constitutively suitable to be used in monitoring the degradation of mechanical properties during operation.

## 4. Conclusions

The effect of the full-year hydrothermal ageing on the flexural and electrical properties of the epoxy and basalt fibre/epoxy composites modified and unmodified with hybrid carbon nanofillers was studied. It was experimentally confirmed that this effect was not relevant for the flexural properties of the epoxy, NC and BFRCs. The relative change of flexural characteristics for all tested materials was within 10–15%. For nano-modified epoxy and FRPs slightly higher effect of absorbed moisture on these characteristics was found, which can be attributed to higher defectiveness (e.g., porosity, the formation of agglomerates etc.).

The results obtained for the glass transition temperature before and after the hydrothermal ageing indirectly indicating that absorbed moisture didn’t induce substantial plasticization of the polymer matrix and degradation of fibre/matrix interface.

Significantly increased electrical conductivity of UD BFRC along fibres due to moisture absorption could be related to the increased electrical conductivity of NC and possible orientation of carbon nanofillers’ particles during the manufacturing of the samples which was evaluated by using modified rules of the mixture.

Satisfactory correlation between the change of electrical conductivity and strain was achieved for all conductive materials. Based on the presented results, it can be concluded that the most environmentally stable and potentially applicable for damage indication is the UD BFRC/NC along fibres.

Thus, the addition of such hybrid nanofillers, at negligible filler content (0.1 wt.% for NC and 0.03 wt.% for BFRC), lead to electrically conductive NC and BFRC system suitably employable in monitoring the degradation of mechanical properties of more complex element during in service operation, e.g., in outdoor conditions.

## Figures and Tables

**Figure 1 polymers-13-00532-f001:**
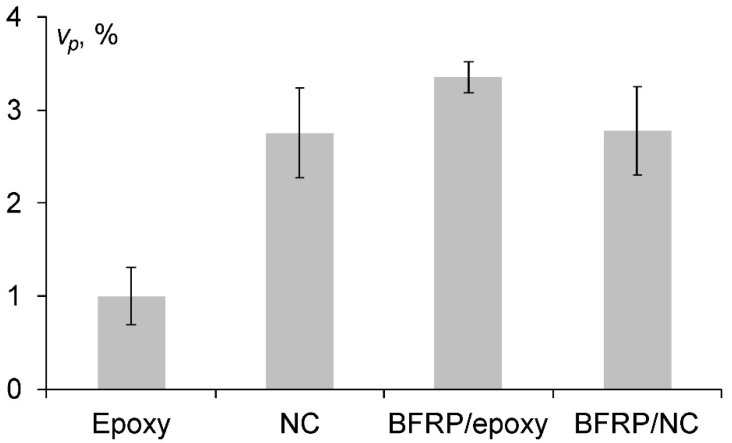
The porosity of the materials investigated (indicated on the graph).

**Figure 2 polymers-13-00532-f002:**
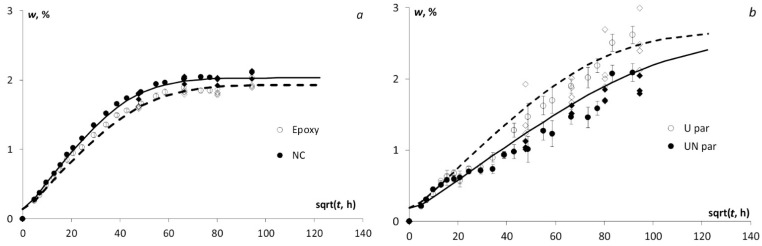
Moisture absorption kinetics of (**a**) the epoxy resin and NC; (**b**) BFRC/epoxy and BFRC/NC parallel to fibres, and (**c**) BFRC/epoxy and BFRC/NC perpendicular to fibres as indicated in the legends and control data after each season for the neat (◊) and nano-modified (♦) materials. Dots—exp. data, lines—calculation by 3D Fick’s model (Equations (3) and (5)).

**Figure 3 polymers-13-00532-f003:**
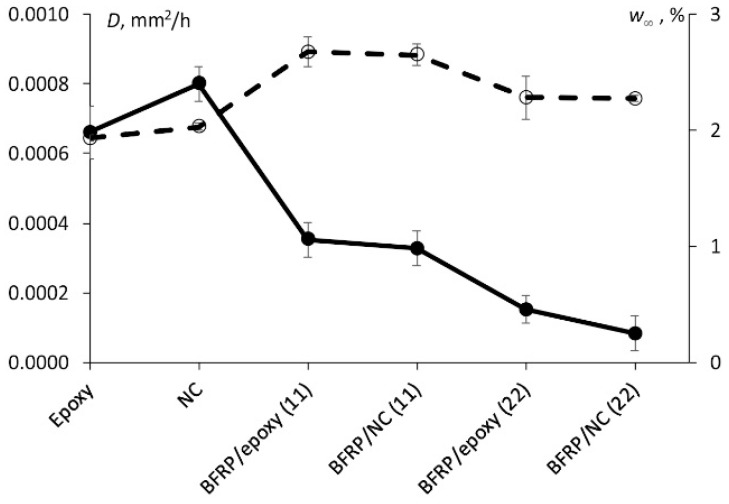
Diffusion coefficient (-●-) and equilibrium water content (-○-) of the materials tested (indicated in the figure).

**Figure 4 polymers-13-00532-f004:**
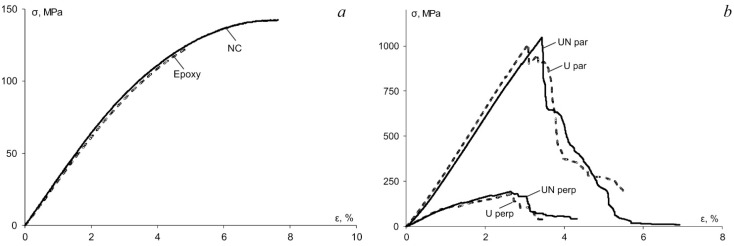
Representative stress-strain curves for (**a**) the epoxy and NC and (**b**) BFRC systems indicated in the legend.

**Figure 5 polymers-13-00532-f005:**
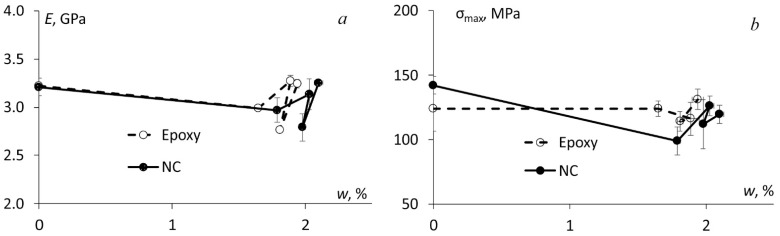
(**a**) Elastic modulus and (**b**) flexural strength of the epoxy and NC vs. absorbed water content measured after each model season as indicated in the legend.

**Figure 6 polymers-13-00532-f006:**
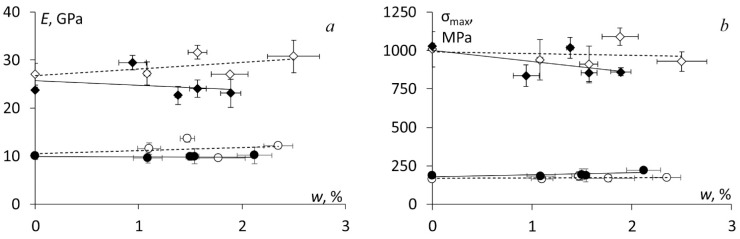
(**a**) Elastic modulus and (**b**) flexural strength of the BFRC systems: U par (◊), UN par (♦), U perp (○), and UN perp (●). Dots – experimental data, lines – linear approximations.

**Figure 7 polymers-13-00532-f007:**
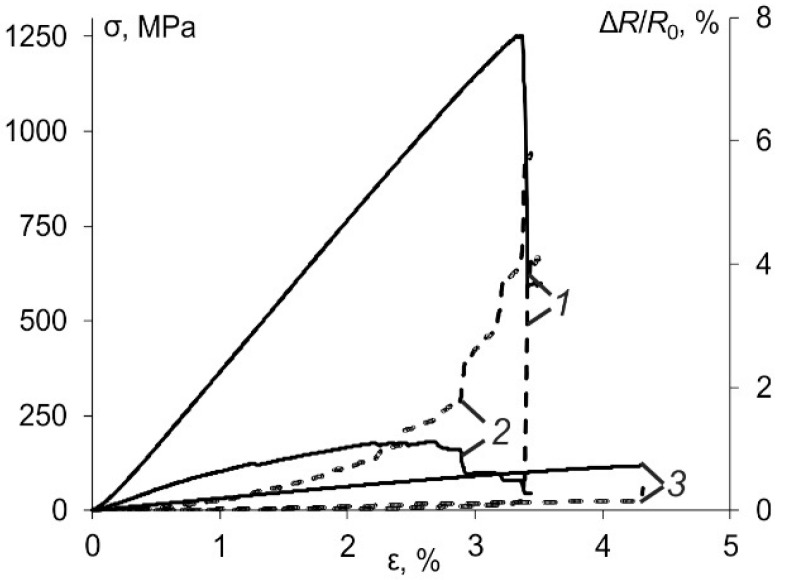
Representative stress-strain curves (solid) and relative change of electrical resistance evaluated by (9) (dashed) for UN par (1), UN perp (2) and NC (3).

**Figure 8 polymers-13-00532-f008:**
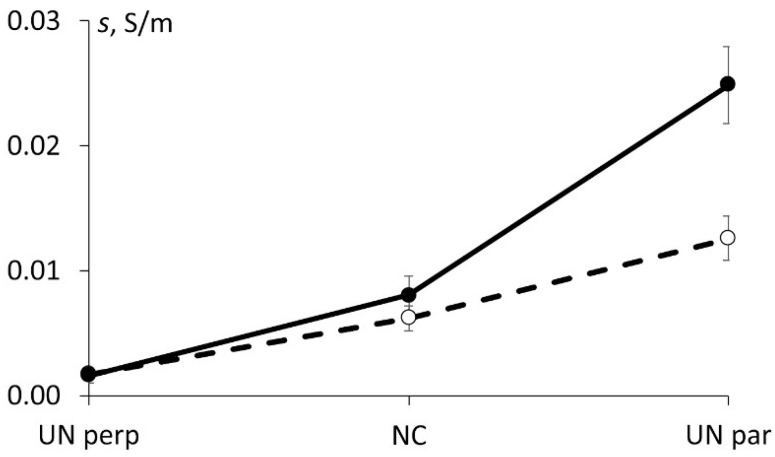
The electrical conductivity of the materials investigated (as indicated) before (-○-) and after (-●-) full-year hydrothermal ageing.

**Figure 9 polymers-13-00532-f009:**
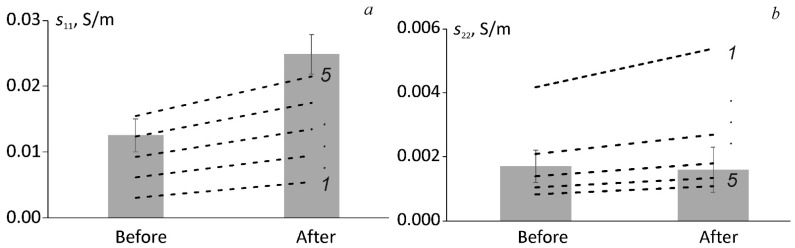
(**a**) The electrical conductivities *s*_11_ and (**b**) *s*_22_ of BFRC along and perpendicular to fibres. Bars—experimental values for UN par and UN perp, lines—evaluation by Equations (11) and (12) for different values of factor n (indicated on the graph).

## Data Availability

The data presented in this study are available on request from the corresponding author.

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
