# Peer review of "Basalt Fibre Composite with Carbon Nanomodified Epoxy Matrix under Hydrothermal Ageing"

_polymers, 2021, doi:10.3390/polym13040532_

Round 1
Reviewer 1 Report
The article is very well designed and written. The flow of the article is well appreciable. References are adequate for the article. Following queries need to be addressed.
- Add some Quantitative critical findings in the abstract.
- Most of the time, authors have written: “described in [11] or reported in [11]”. It advised adding relevant information for the benefit of the reader.
Reviewer 2 Report
- Pag 1 line 42 change [7,8,9] for [7-9]
- 2. Materials and Methods, include information about purification materials or used as received
- Pag 2, line 92 revise if is correct this information “[0]8”
- Manuscript needs revision by native English speaking (are was the mass?)
- Manuscript has some interesting results but has no discussion, include it for all figures
